# The Effects of Short-Term Exposure to pH Reduction on the Behavioral and Physiological Parameters of Juvenile Black Rockfish (*Sebastes schlegelii*)

**DOI:** 10.3390/biology12060876

**Published:** 2023-06-17

**Authors:** Haixia Li, Jia Zhang, Xiaoyu Ge, Songmeng Chen, Zhen Ma

**Affiliations:** 1Key Laboratory of Environment Controlled Aquaculture, Ministry of Education, Dalian 116023, China; 2College of Marine Technology and Environment, Dalian Ocean University, Dalian 116023, China

**Keywords:** short-term stress, pH reduction, behavioral performance, physiological response, *Sebastes schlegelii*

## Abstract

**Simple Summary:**

The reduction in seawater pH as a consequence of ocean acidification has resulted in a considerable challenge for the adaptation of fish. The effects of pH fluctuations in coastal regions on offshore fish are not fully understood in terms of their behavioral and physiological responses. In the present study, we aimed to examine the consequences of a brief period of reduced pH levels on the behavioral and physiological performance of juvenile black rockfish (*Sebastes schlegelii*). The findings revealed that juvenile black rockfish could withstand changes in pH, but there was a significant impact on their activity state and metabolic levels. The present study will enhance our understanding of the behavioral and physiological responses of costal fish (the black rockfish, in this case) to acidic conditions.

**Abstract:**

Coastal areas are subject to greater pH fluctuation and more rapid pH decline as a result of both natural and anthropogenic influences in contrast to open ocean environments. Such variations in pH have the potential to pose a threat to the survival and physiological function of offshore fishes. With the aim of evaluating the impact of short-term pH reduction on the behavioral performance and physiological response of costal fish, the black rockfish (*Sebastes schlegelii*), one of the principal stock-enhanced species, was examined. In the present study, juveniles of the black rockfish with a mean body length of 6.9 ± 0.3 cm and weight of 8.5 ± 0.5 g were exposed to a series of pHs, 7.0, 7.2, 7.4, 7.6, 7.8, and normal seawater (pH 8.0) for 96 h. At the predetermined time points post-exposure (i.e., 0, 12, 24, 48, and 96 h), fish movement behavior was recorded and the specimens were sampled to assess their physiological responses. The results indicate that the lowered pH environment (pH 7.0–7.8) elicited a significant increase in highly mobile behavior, a decrease in immobile behavior, and a significant rise in the metabolic levels of the black rockfish juveniles. Specifically, carbohydrate metabolism was significantly elevated in the pH 7.2 and 7.4 treatments, while lipid metabolism was significantly increased in the pH 7.0, 7.4, and 7.8 treatments. The results of the present study indicate that short-term reductions in pH could ramp up boldness and boost energy expenditure in the black rockfish juveniles, leading to an increased metabolic cost. Additionally, the present investigation revealed that the black rockfish juveniles were capable of adapting to a short-term pH reduction. The findings may provide insight into the underlying physiological mechanisms that govern fish responses to potential decreases in seawater pH in the future.

## 1. Introduction

Ocean acidification (OA) is a problem that poses a serious threat to the stability and function of marine ecosystems [1]. It is projected that surface seawater pH will decrease by 0.3 to 0.4 units by the end of the 21st century and by 0.7 to 0.8 units by 2300 [2,3]. A particular concern is the swift fluctuation of pH in coastal areas, which are the primary habitat for many marine organisms [4]. The pH in these areas undergoes significant fluctuations, often by more than one unit, owing to seasonal cycles of biological production, river flow, upwelling, or anthropogenic factors compounded by the impact of general OA [5,6,7,8,9,10]. The pH of coastal waters is highly heterogeneous with daily coastal pH fluctuations of 0.2 to 0.4 units [11,12], surpassing that of open oceans [13]. For example, one study indicated that the pH of estuaries can range from 7.45 to 8.10 annually [14]. Long-term observations suggest that coastal areas experience faster pH decline than open oceans [15]. Thus, it is imperative to investigate acute acidic stress in coastal areas, characterized by large daily pH variations.

Rapid changes in seawater pH have the potential to directly or indirectly affect the growth and development of marine organisms [16]. Increasing studies have demonstrated that even fish with a relatively well-developed acid-base balance regulation system would be negatively impacted by decreases in pH, particularly during the early developmental stages [17,18]. A lower pH has been found to harm the olfactory sensitivity and capacity of fish to distinguish between odors from their habitat [19], to diminish their ability to detect sounds [20], decrease their alertness [21] and ability to evade predation [22], and disrupt the function of their neurological systems [23], ultimately reducing their adaptability and altering fish abundance and community biodiversity.

The response of fish to external stimuli manifests as a change in their behavior, which plays a vital role in their survival and adaptability to environmental conditions [24]. Fish behavior can serve as an indicator of changes in the ecological environment. Fish exhibit a range of behavioral strategies to cope with short-term environmental changes and alleviate discomfort, thereby facilitating adaptive regulation [25]. However, when the environmental stresses exceed a certain threshold, fish experience compromised physiological functions, leading to aberrant behavior [26,27,28,29]. Thus, behavioral monitoring represents a non-invasive, rapid, and intuitive method for linking ecological changes to fish physiological functions and determining their survival status [30]. Recent studies on fish behavior and seawater ecological changes have focused on the impact of decreasing seawater pH on fish behavior [31,32]. Studies have revealed that acidic environments could enhance swimming activity and boldness levels in fish [33], increase anxiety levels [34], impair olfactory-dependent avoidance behavior [35], and affect behavioral lateralization in the brain [36]. Nevertheless, other studies have reported weak behavioral effects [37] or reduced activity [38], which suggests that differences in fish behavior may be influenced by a variety of factors, such as population, habitat, pH, and duration of experiments. Under acute stress resulting from a decrease in pH, fish demonstrate abnormal behaviors and adaptation regulation, although the underlying character of these behaviors remains unclear.

Exposure to acidified seawater directly affects the adaptive load of fish. A continuous decrease in pH triggers a significant physiological stress response in fish, leading to an increase in glucocorticoid levels, particularly cortisol, mediated through the hypothalamic-pituitary-interrenal (HPI) axis [39,40]. This stress response further results in increased gas exchange and glucose concentration in fish [41]. Maintaining the biological functioning of fish within the ideal physiological range in suboptimal environments is energetically costly. Fish increase their carbohydrate and lipid metabolism levels to meet the higher energy demands [42]. However, when the acidic environment exceeds the range that fish can tolerate, they consume excess energy and inhibit the function of their specific and non-specific immune defense system [43]. This decrease in resistance to disease may even result in fish death. Therefore, evaluating the influence of seawater acidification on fish physiological responses is crucial to gain a better understanding of how fish adjust their physiological status to the degree of acidification.

The black rockfish (*Sebastes schlegelii*) is a viviparous teleost species that primarily inhabits near-shore reefs. This species is found abundant in the northwestern Pacific Ocean, particularly in China, Korea, and Japan, and has considerable commercial importance [44,45]. Unfortunately, the wild population of the black rockfish has drastically declined due to overfishing, offshore development activities, and massive pollutant releases [46,47]. Restocking the black rockfish has proven to be an effective strategy to revive the resource [48]. More than 1.5 million juveniles have been released every year over the past five years [49,50]. However, the fluctuating environmental factors in the habitat of the black rockfish, particularly the variable seawater pH driven by both natural and anthropogenic factors, might undermine the efficacy of this strategy. Consequently, the present study endeavors to investigate the behavioral and physiological responses of the black rockfish to short-term pH stress to enhance the resource management of the species, and provide insights into the physiological mechanisms of fish response to future seawater pH reduction.

## 2. Materials and Methods

### 2.1. Animals

In the present study, 375 black rockfish juveniles were sourced from a local commercial hatchery and subjected to an acclimatization period of 14 days in a 750 L polypropylene plastic cylinder. The seawater used for both acclimatization and experimental purposes was naturally sand filtered at the Key laboratory of environment-controlled aquaculture (AET) of Dalian Ocean University. Throughout the acclimatization period, the temperature ranged from 19 °C to 21 °C, salinity was 31, and the natural photoperiod was applied. Additionally, the fish were fed twice daily with commercial dry pellets containing 48% proteins and 9% lipids, at a rate of 2% of body weight.

### 2.2. Experimental Design

The experiments commenced on the 15 January 2021 and ended on the 16 March 2021. Natural seawater with a pH of 8.0 was used as the control, and five pHs, 7.0, 7.2, 7.4, 7.6, and 7.8 were tested in triplicate. These pH values were categorized into the higher-pH treatments (i.e., pH 7.6 and 7.8) and the lower-pH treatments (i.e., pH 7.0, 7.2, and 7.4) based on predictions concerning future pH fluctuations. In total, 360 juveniles of the black rockfish (6.9 ± 0.3 cm in body length, and 8.5 ± 0.5 g in weight) were randomly allocated into 18 white polypropylene plastic boxes (0.67 m × 0.50 m × 0.37 m, 0.2 m in water depth), each with 20 juveniles. The pH values were prepared with HCl 18% (5.87 M) and measured using a YSI multi-parameter water quality analyzer, and the pH value of the experimental water was adjusted twice daily at 9:00 and 17:00. Fish mortality was recorded daily, and survival rates were calculated. The fish were not fed to maintain water quality during the 96-h experiment.

### 2.3. Behavioral Analysis

A video camera (HIK-DS-2CD3T35D-I5) was set up above each experimental box to record fish behavior. At each time point, i.e., 0, 12, 24, 48, and 96 h post exposure, fish behavior was recorded for 10 min. The videos were analyzed using Noldus EthoVision XT (version 12.0; Noldus Information Technology, Wageningen, The Netherlands) in conjunction with an all-occurrence recording method. The analysis was conducted by dynamically subtracting the background noise of water surface light spots and fluctuations using the smooth (Lowess) method. For video analysis, eight fish were individually tracked and identified for each experimental box [51]. Three behavioral parameters including locomotor activity, activity state, and proximity were measured. Locomotor activity was characterized by velocity, maximum acceleration, and angular velocity. Velocity (cm/s) was calculated as the average velocity of the center point of the observed sample during the observation time. Maximum acceleration (cm/s^2^) was calculated as the maximum value of the time difference divided by the velocity difference of two consecutive frames of the observed object. Angular velocity (º/s) was calculated as the average of the change in direction of the object centroid in two consecutive frames per unit time. Activity state was classified into three categories based on the magnitude of total pixels varied per unit time, i.e., highly mobile, mobile, or immobile. Fish were classified as being in the highly mobile state if the total pixels varied by more than 60% per unit time, while the mobile state was assigned if the total pixels varied by more than 20% per unit time. Conversely, the immobile state was designated when the percentage of changed pixels did not exceed the aforementioned thresholds. These parameters served as indicators of the activity and position of the juvenile fish. In addition, proximity was used to evaluate the social activity of the fish, which could be indicative of their schooling behavior. The proximity state was defined as the distance between the target object, and the center point of other objects is less than the body length of the fish.

### 2.4. Sampling and Physiological Measurement

Immediately after the behavioral study assessment, three juveniles were randomly selected from each experimental box for physiological measurements. The fish were anesthetized with eugenol (80 mg/L) for 10–20 s and their liver tissue were carefully excised and rinsed with ice-cold 0.86% NaCl solution to minimize damage. Liver tissue was used for experimentation due to its vital role in fish metabolism and adaptation to environmental changes. The extracted livers were frozen in liquid nitrogen and stored at −80 °C until the analysis. The samples were mechanically homogenized with ice-cold saline in an ice-water bath and afterwards centrifuged at 2500 rpm for 10 min at 4 °C to produce the supernatant for subsequent analysis.

The oxidative stress status of the black rockfish juveniles was determined by measuring the activity of total superoxide dismutase (T-SOD), catalase (CAT), and glutathione peroxidase (GSH-PX). The calcium and phosphorus balance of fish was evaluated using alkaline phosphatase (AKP), an immunological indicator. The metabolic status of carbohydrates and lipids in fish was analyzed by assessing glucose (GLU) and triglyceride (TG) physiological parameters. Cortisol levels, an indicator of the immediate stress response of fish, were also measured. Commercially available kits from the Nanjing Jiancheng Institute of Biological Engineering (Nanjing, China) were employed to quantify the activity of these physiological parameters, along with cortisol levels that were measured using an ELISA kit.

### 2.5. Statistical Analysis

Statistical analyses were carried out using SPSS 26.0. All values were presented as means ± standard error. The data were transformed through the application of appropriate techniques, such as rank transformation, arcsine transformation, or logarithmic transformation in cases where violations of the assumptions of normality and homogeneity were detected. In order to evaluate the impact of pH, time, and their interaction on the behavioral parameters, Generalized linear mixed models (GLMMs) were used to fit the behavioral data. Fixed coefficients and estimates were reported for all effects under investigation. Meanwhile, the separate and interaction effects of pH and time on the physiological data were investigated through Two-way ANOVA. Tukey’s post hoc test was then performed to determine the significant differences between the means of the factors for physiological data.

## 3. Results

### 3.1. Behavior Analysis

#### 3.1.1. Locomotor Activity

The results show statistically significant interaction effects between pH (pH 7.2–7.8) and time (*p* < 0.05, Table 1). At the 12-h time point, the velocity of the lower-pH treatments (pH 7.0, 7.2, and 7.4) was significantly lower than the control group (*p* < 0.05), while the higher-pH treatments (pH 7.6 and 7.8) did not differ significantly (*p* > 0.05, as shown in Table 2). At 24 h, the velocity of the pH 7.2 and 7.8 treatments showed significant increases compared to the control (*p* < 0.05). All treatments demonstrated significantly higher velocity than the control within 48 h (*p* < 0.05), except for the pH 7.0 treatment at 96 h, which had significantly lower velocity than the control (*p* < 0.05).

The findings suggest that the interaction between time and pH did not account for the variation in maximum acceleration (*p* > 0.05). However, a significant reduction in maximum acceleration was observed in response to decreased pH, with significant effects seen in the pH 7.2, 7.6, and 7.8 treatments (*p* < 0.05, Table 3). Additionally, compared to the control group, the pH 7.0 and 7.4 treatments exhibited significantly lower maximum acceleration at 12 h (*p* < 0.05, Table 2). At 24 h, all treatments demonstrated significant differences from the control (*p* < 0.05), but not at 48 h (*p* > 0.05). At 96 h, the pH 7.4, 7.6, and 7.8 treatments showed significantly lower maximum acceleration than the pH 7.2 treatment (*p* < 0.05) but did not differ significantly from the control (*p* > 0.05).

The findings indicated that there was not a significant interaction between pH and time on the variation in angular velocity (*p* > 0.05, Table 4). However, at 96 h, the pH 7.8 treatment showed a significantly lower angular velocity than the control group, while fish in the pH 7.0 treatment exhibited a significantly higher angular velocity than in the other treatments (*p* < 0.05, Table 2). No significant difference was observed at any other time point (*p* > 0.05).

#### 3.1.2. Activity State

The effects of fixed factors, including time, pH, and their interactions, were analyzed to understand their impact on highly mobile behavior. No significant effects were observed in the higher-pH treatments (pH 7.6 and 7.8) (*p* > 0.05, Table 5). At 12 h and 24 h, the pH 7.0, 7.2, 7.6, and 7.8 treatments exhibited significantly higher highly mobile behavior than the control group (*p* < 0.05, Table 6). Within 48 h, fish in the pH 7.2, 7.4, and 7.8 treatments exhibited significantly higher highly mobile behavior than in the control (*p* < 0.05). At 96 h, all treatments displayed significantly higher highly mobile behavior than the control. Furthermore, the higher-pH treatments exhibited significantly lower highly mobile behavior than the pH 7.0 and 7.2 treatments (*p* < 0.05), but no significant difference was observed between the higher-pH treatments (pH 7.6 and 7.8) and the pH 7.4 treatment (*p* > 0.05).

The statistical analysis revealed that none of these fixed factors had a significant effect on mobile behavior (*p* > 0.05, Table 7). However, at 12 h, fish in all treatments exhibited a significant reduction in mobile behavior compared to the control (*p* < 0.05, Table 6). Among the treatments, the pH 7.0 group exhibited the lowest mobile behavior during 24 h, which was significantly different from the other treatments and the control (*p* < 0.05). No significant differences were observed in mobile behavior at 48 h (*p* > 0.05). At 96 h, the pH 7.6 treatment had a significantly higher mobile behavior than the pH 7.0 and 7.4 treatments and the control (*p* < 0.05), while no other significant differences were detected (*p* > 0.05).

The statistical analysis revealed the significant effects of fixed factors on the immobile behavior of the black rockfish juveniles (*p* < 0.05). However, pH 7.2, 7.6, and 7.8 treatments did not significantly affect immobile behavior (*p* > 0.05, Table 8). At 12 h, fish in the pH 7.0 and 7.4 treatments showed significantly higher immobile behavior than the higher-pH treatments (pH 7.6 and 7.8) and the control (*p* < 0.05, Table 6). At 24 h, the pH 7.8 treatment exhibited significantly lower immobile behavior than the other treatments and the control, while the pH 7.0 treatment had significantly higher immobile behavior than other treatments and the control (*p* < 0.05). Additionally, at 48 h, the higher-pH treatments had significantly lower immobile behavior than the control (*p* < 0.05). All treatments showed significantly lower immobile behavior than the control within 96 h, of which the pH 7.0 treatment exhibited the lowest immobile behavior (*p* < 0.05). No other significant differences were detected among the treatments (*p* > 0.05).

#### 3.1.3. Proximity

According to the results presented in Table 9, the proximity of the black rockfish juveniles had a significant effect (*p* < 0.05) in time, while the pH and its interaction did not have a significant effect (*p* > 0.05). Specifically, within 12 h, only the pH 7.4 treatment showed a significant increase in proximity compared to the control (*p* > 0.05, Table 10). At 24 h, all treatments had significantly lower proximity than the control, of which fish in the pH 7.2, 7.4, and 7.8 treatments showed significantly lower proximity than in the pH 7.0 and 7.6 treatments (*p* < 0.05). At 48 h, the pH 7.0 and 7.2 treatments had significantly lower proximity than the pH 7.4 and 7.6 treatments (*p* < 0.05), but the proximity of all treatment groups was not significantly different from the control (*p* > 0.05). Notably, at 96 h, the pH 7.4 treatment exhibited significantly higher proximity than the pH 7.0 and 7.2 treatments, and the control (*p* < 0.05).

### 3.2. Physiological Response

#### 3.2.1. Immune Function

The result shows that the pH and time interaction significantly affected the immune system of the black rockfish juveniles (*p* < 0.05, Table 11). At 12 h, the T-SOD activity of the pH 7.0, 7.4, and 7.8 treatments was significantly higher than the control (*p* < 0.05). At 48 h, the T-SOD activity of the pH 7.2 and 7.6 treatments was significantly lower than the control (*p* < 0.05), while no significant differences were observed in the other treatments (*p* > 0.05, Figure 1a). CAT activity did not differ significantly between all treatments and the control (*p* > 0.05, Figure 1b). During 24 h, GSH-PX activity was significantly lower in the pH 7.2 and 7.4 treatments than the control (*p* < 0.05), with no other significant differences observed (*p* > 0.05, Figure 1c). Figure 1d shows that the pH 7.8 treatment had significantly higher AKP activity than the control at 12 h and 24 h (*p* < 0.05). At 96 h, fish in the pH 7.2 and 7.4 treatments also showed significantly higher activity than in the control (*p* < 0.05), while no other significant differences were observed (*p* > 0.05).

#### 3.2.2. Metabolic Level

The metabolic responses of the black rockfish juveniles are significantly affected by the interaction between pH and time (*p* < 0.05, Table 12). GLU levels of fish were significantly higher in the pH 7.4 treatment than the control at 12 h (*p* < 0.05, Figure 2a), while the pH 7.2 treatment had significantly higher levels than the control at 24 h (*p* < 0.05), but no other significant differences were observed (*p* > 0.05). TG level of fish showed significant differences in the pH 7.0 and 7.8 treatments compared to the control at 12 h, with higher levels observed in these groups (*p* < 0.05, Figure 2b). Fish in the pH 7.2 treatment compared to the control at 48 h had lower TG levels (*p* < 0.05, Figure 2b). No significant differences were observed in other treatments (*p* > 0.05).

#### 3.2.3. Stress Response

The interaction between pH and time was found to have a significant effect on the stress response of the black rockfish juveniles (*p* < 0.05, Table 13). Cortisol levels of fish were observed to be significantly higher in the pH 7.0 and 7.4 treatments than in the control over a period of 24 h (*p* < 0.05, Figure 3), while no significant differences were observed in the other treatments (*p* > 0.05).

## 4. Discussion

The rapid fluctuations in pH levels have the potential to directly or indirectly influence the physiological and behavioral traits of marine organisms. The implications of pH alterations on fish behavior remain a subject of ongoing debate. Our primary research findings provided evidence that a temporary reduction in pH levels resulted in an increased activity state in the fish, accompanied by notable enhancements in carbohydrate and lipid metabolism. Juveniles could tolerate certain reductions in pH levels, and their physiological responses can return to baseline levels within a short period of time.

The study revealed that the black rockfish juveniles had a significant increase in both velocity and acceleration in the higher-pH environments (pH 7.6 and 7.8), but this effect decreased as the pH level declined. This finding is contrary to a previous study by Sundin et al. [52], which could be attributed to the fact that the black rockfish is a rocky species that inhabits waters with large seasonal and interannual pH fluctuations, resulting in their reduced sensitivity to weak pH fluctuations [6,53,54]. A few other studies also found that the black rockfish juveniles can adapt to sudden pH fluctuations by enhancing their swimming velocity [55,56]. In contrast, the velocity of juvenile the black rockfish was not significantly affected by the pH reduction to 7.0, which may be due to the delay in the sensory ability of the fish when the pH decreased to a certain level, thereby reducing the ability of the fish to respond rapidly [57]. Exposure to seawater with a decreased pH may result in less sensitive olfactory receptors and impaired processing at the neurological level, leading to significant alterations in the lateralization intensity and direction of the fish [58,59]. However, the present study found no significant difference in the black rockfish’s directional awareness after short-term exposure to a reduced pH environment (i.e., pH 7.0–7.8), contrary to previous studies [23,60,61]. The exposure time could be a contributing factor, as it might not be sufficient to have a significant impact on the lateralization of the black rockfish juveniles [62].

The present study found that as pH decreased, the black rockfish juveniles showed a significant increase in highly mobile behavior and a decrease in immobile behavior. These findings are consistent with previous studies [33,57,63], suggesting that reduced pH makes fish bolder and more anxious, potentially increasing their predation rates and susceptibility to being caught by fishing gear [64,65]. Furthermore, the present study found that carbohydrate and lipid metabolism levels of the black rockfish juveniles were significantly elevated in the lower-pH treatments (pH 7.0, 7.2, and 7.4). This is consistent with previous research indicating that decreased pH causes fish to become more highly mobile, resulting in increased consumption of carbohydrates and lipids and significantly raising metabolic levels [66]. The immobile behavior of the black rockfish juveniles was significantly reduced after 48 h in the higher-pH treatments (pH 7.6 and 7.8), while mobile behavior showed an increasing trend, suggesting that the impact of pH reduction on fish behavior is time dependent. Minor pH decreases had no significant effect on the active state of the black rockfish juveniles and their metabolic levels.

The shoaling behavior of fish can provide insights into their survival strategies both as individuals and as a group when facing environmental threats. The present study indicated that reduced pH did not significantly affect the proximity of the black rockfish, but it did reduce shoaling behavior and led to population dispersion at the end of the experiment in the lower-pH treatment (pH 7.4). This result contradicts previous studies, but as exposure to low pH was brief, short-term pH fluctuations may have limited impact on shoaling behavior [27,36]. However, the black rockfish’s tendency to disperse in low pH environments implies that prolonged exposure may diminish their ability to adapt to pH change [67].

Fish may suffer from acidosis due to the formation of insoluble compounds with proteins under high H^+^ concentrations, which could result in tissue and organ damage or even death. The present study suggests that the black rockfish juveniles may tolerate greater pH change, as none of the pH treatments caused mortality at the end of the experiment. These findings demonstrate that fish’s robust resilience to acidic conditions [68].

Fish have unique antioxidant and detoxifying defense mechanisms that help them survive under stressful conditions [69,70]. A previous study revealed that exposure to reduced pH environments can induce oxidative stress in fish and that the severity and duration of exposure are linked to these responses [71]. In the current work, the black rockfish juveniles exposed to varying levels of pH stress experienced oxidative stress, which returned to baseline after the experiment. The present study further demonstrated that the black rockfish juveniles can withstand short-term pH stress that causes acidic stress. The activity of AKP, a crucial enzyme in the immune and cellular oxygen-carrying system of fish, was significantly greater in the pH 7.2 and 7.4 treatments than in the control, consistent with a previous study [72]. These results suggest that lower-pH conditions (pH 7.2 and 7.4) may necessitate increased AKP activity to maintain regular phosphate group transfer and temporarily stimulate the immune system of the black rockfish [73], but further testing is required to assess potential immune system damage.

Fish have evolved specialized mechanisms to adapt to extreme changes in seawater, including metabolic and immune responses that regulate homeostasis through ionic and osmotic regulation [74]. However, environmental changes can increase the metabolic cost of fish [75,76], leading them to enhance their carbohydrate and lipid metabolism to cope with the high energy demands during acidification stress [77,78,79]. The current study investigated the impact of pH stress on the black rockfish juveniles. The findings indicated that TG levels were significantly elevated in both the higher-pH (pH 7.8) and the lower-pH (pH 7.0 and 7.4) treatments, while GLU levels were significantly higher only in the lower-pH treatments (pH 7.2 and 7.4) and not significantly different in the higher-pH treatments (pH 7.6 and 7.8). These results may be related to the variable temperature lifestyle of fish, resulting in decreased sensitivity to carbohydrate metabolism and reduced glucose utilization [80].

Cortisol is a reliable stress indicator in fish [81]. The black rockfish juveniles displayed signs of stress under the lower-pH treatments (pH 7.0 and 7.4) for 24 h, but not in the higher-pH treatments (pH 7.6 and 7.8). This suggests that minor pH changes are insufficient to cause stress in the black rockfish juveniles [82], further demonstrating their insensitivity to minor pH fluctuations in seawater. Additionally, the results showed that cortisol levels in all pH treatment groups returned to baseline by the end of the experiment, indicating that the black rockfish juveniles may quickly adapt to the lower pH environment of saltwater. These findings contribute to our understanding of the physiological responses of the black rockfish juveniles to pH stress and highlight the remarkable resilience of these fish in acidic conditions.

## 5. Conclusions

In the present study, we investigated the impact of short-term pH reduction stress on the behavioral and physiological responses of black rockfish juveniles. Our findings suggest that the black rockfish juveniles exhibited increased highly mobile behavior and decreased immobile behavior under lower pH conditions. The higher lipid and carbohydrate metabolism observed in the black rockfish juveniles are indicative of how lower pH conditions affect their activity state. Specifically, carbohydrate metabolism was significantly enhanced in the lower-pH treatments (pH 7.2 and 7.4), whereas lipid metabolism was significantly elevated in both the higher-pH (pH 7.8) and lower-pH treatments (pH 7.0 and 7.4). Additionally, the present study showed that the black rockfish juveniles are capable of tolerating pH changes and adapting to seawater environments with reduced pH over a short period. Overall, the present study provides data that serve as a theoretical basis for understanding the underlying physiological mechanisms of fish responses to future seawater pH reduction.

## Figures and Tables

**Figure 1 biology-12-00876-f001:**
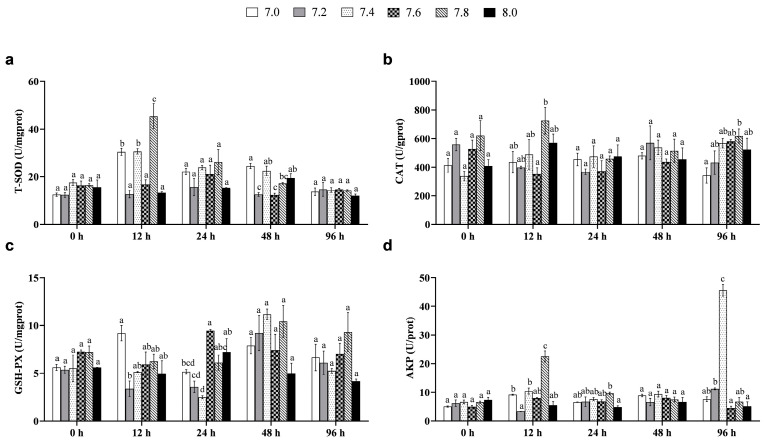
Immune function in the black rockfish juveniles exposed to short-term pH reduction for 96 h. (**a**) Total superoxide dismutase (T-SOD); (**b**) Catalase (CAT); (**c**) Glutathione peroxidase (GSH-PX); (**d**) Alkaline phosphatase (AKP). Data with different letters differ significantly (*p* < 0.05) among groups.

**Figure 2 biology-12-00876-f002:**
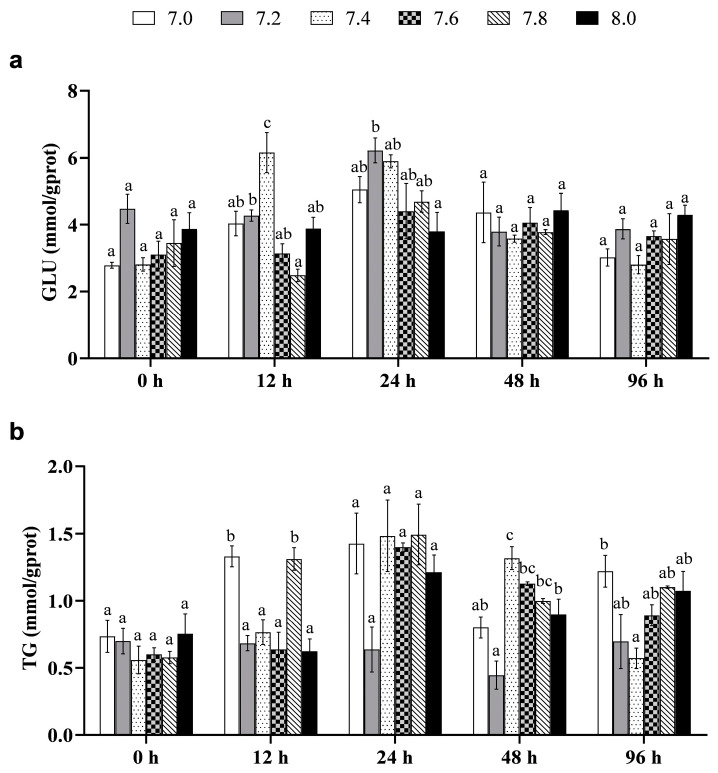
Metabolic responses in the black rockfish juveniles exposed to short-term pH reduction for 96 h. (**a**) Glucose (GLU); (**b**) Triglyceride (TG). Data with different letters differ significantly (*p* < 0.05) among groups.

**Figure 3 biology-12-00876-f003:**
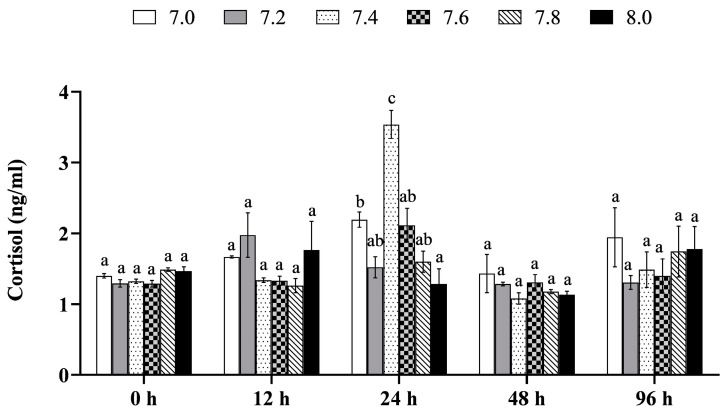
Stress response in the black rockfish juveniles exposed to short-term pH reduction for 96 h. Data with different letters differ significantly (*p* < 0.05) among groups.

**Table 1 biology-12-00876-t001:** Effects of pH, time, and their interaction on swimming velocity in the black rockfish juveniles using GLMMs.

	Explanatory Variables	Coefficient	SE	*t* Value	*p*-Value
Velocity (cm/s)	pH 7.8	−0.624	0.375	−1.662	0.098
	pH 7.6	−0.545	0.375	−1.452	0.148
	pH 7.4	−1.243	0.375	−3.314	**0.001**
	pH 7.2	−0.941	0.375	−2.510	**0.013**
	pH 7.0	−0.615	0.375	−1.639	0.103
	Time	−0.647	0.080	−8.097	**<0.001**
	Time * [pH 7.8]	0.451	0.113	3.985	**<0.001**
	Time * [pH 7.6]	0.383	0.113	3.384	**0.001**
	Time * [pH 7.4]	0.382	0.113	3.382	**0.001**
	Time * [pH 7.2]	0.349	0.113	3.086	**0.002**
	Time * [pH 7.0]	0.067	0.113	0.590	0.556

Significant effects are shown in bold. Time * pH indicates the effect of the interaction of time and pH on the velocity.

**Table 2 biology-12-00876-t002:** Effects of short-term exposure to pH reduction on locomotor activity in the black rockfish juveniles.

	Time	pH					
		8.0	7.8	7.6	7.4	7.2	7.0
Velocity (cm/s)	0 h	0.95 ± 0.14	0.74 ± 0.16	0.89 ± 0.17	0.71 ± 0.11	0.62 ± 0.14	0.76 ± 0.14
	12 h	0.73 ± 0.16 ^a^	0.01 ± 0.06 ^ab^	0.60 ± 0.11 ^a^	−1.54 ± 0.06 ^b^	−0.66 ± 0.03 ^b^	−0.73 ± 0.02 ^b^
	24 h	−0.08 ± 0.05 ^a^	2.19 ± 0.11 ^b^	0.90 ± 0.16 ^ab^	1.16 ± 0.17 ^ab^	1.17 ± 0.10 ^b^	−0.04 ± 0.05 ^a^
	48 h	−0.92 ± 0.04 ^a^	0.31 ± 0.07 ^e^	−0.03 ± 0.03 ^d^	−0.38 ± 0.06 ^c^	−0.36 ± 0.04 ^c^	−0.65 ± 0.07 ^b^
	96 h	−1.46 ± 0.08 ^b^	−0.39 ± 0.04 ^d^	−0.12 ± 0.06 ^d^	−1.20 ± 0.04 ^bc^	−1.02 ± 0.06 ^c^	−2.19 ± 0.11 ^a^
Maximum Acceleration (cm/s^2^)	0 h	0.35 ± 0.15	0.49 ± 0.05	0.56 ± 0.07	0.36 ± 0.08	0.42 ± 0.13	0.22 ± 0.08
	12 h	−0.66 ± 0.25 ^a^	−0.05 ± 0.17 ^ab^	0.30 ± 0.20 ^ab^	−2.19 ± 0.11 ^b^	2.18 ± 0.12 ^a^	−0.70 ± 0.15 ^b^
	24 h	−1.44 ± 0.13 ^a^	0.97 ± 0.07 ^c^	−0.55 ± 0.27 ^b^	0.56 ± 0.17 ^c^	1.66 ± 0.04 ^d^	−0.54 ± 0.12 ^b^
	48 h	−0.56 ± 0.16	−0.84 ± 0.20	−0.33 ± 0.14	−0.69 ± 0.18	−0.60 ± 0.08	−0.46 ± 0.11
	96 h	0.42 ± 0.18 ^ab^	−0.06 ± 0.22 ^b^	−0.30 ± 0.28 ^b^	−0.96 ± 0.06 ^b^	1.29 ± 0.05 ^a^	1.15 ± 0.14 ^ab^
Angular Velocity (º/s)	0 h	−0.10 ± 0.14	−0.16 ± 0.14	−0.19 ± 0.06	−0.22 ± 0.10	−0.24 ± 0.21	−0.05 ± 0.12
	12 h	0.58 ± 0.14	0.23 ± 0.40	0.12 ± 0.32	0.35 ± 0.22	0.17 ± 0.18	0.46 ± 0.27
	24 h	−0.23 ± 0.20	−0.77 ± 0.17	−0.09 ± 0.22	−0.72 ± 0.45	−0.60 ± 0.28	0.03 ± 0.20
	48 h	−0.83 ± 0.20	−0.67 ± 0.17	−0.92 ± 0.23	−1.17 ± 0.55	−1.48 ± 0.25	−1.09 ± 0.19
	96 h	1.63 ± 0.18 ^ab^	0.80 ± 0.07 ^c^	1.17 ± 0.19 ^bc^	1.01 ± 0.16 ^bc^	1.13 ± 0.13 ^bc^	1.86 ± 0.19 ^a^

Different letters indicate significant differences between groups within the same row, and same letters indicate no significant difference between groups within the same row. The values indicate estimates ± standard error using GLMMs.

**Table 3 biology-12-00876-t003:** Effects of short-term pH reduction on locomotor activity in the black rockfish juveniles using GLMMs.

	Explanatory Variables	Coefficient	SE	*t* Value	*p*-Value
Maximum Acceleration (cm/s^2^)	pH 7.8	1.120	0.445	2.515	**0.013**
	pH 7.6	1.091	0.445	2.449	**0.015**
	pH 7.4	0.206	0.445	0.461	0.645
	pH 7.2	1.751	0.445	3.931	**<0.001**
	pH 7.0	−0.243	0.445	−0.545	0.587
	Time	0.025	0.095	0.259	0.796
	Time * [pH 7.8]	−0.214	0.134	−1.597	0.112
	Time * [pH 7.6]	−0.260	0.134	−1.934	0.054
	Time * [pH 7.4]	−0.137	0.134	−1.023	0.307
	Time * [pH 7.2]	−0.128	0.134	−0.953	0.342
	Time * [pH 7.0]	0.186	0.134	1.382	0.168

Significant effects are shown in bold. Time * pH indicates the effect of the interaction of time and pH on the maximum acceleration.

**Table 4 biology-12-00876-t004:** Effects of pH, time, and their interaction on angular velocity in the black rockfish juveniles using GLMMs.

	Explanatory Variables	Coefficient	SE	*t* Value	*p*-Value
Angular Velocity (º/s)	pH 7.8	0.008	0.299	0.028	0.978
	pH 7.6	−0.114	0.299	−0.381	0.704
	pH 7.4	−0.007	0.299	−0.024	0.981
	pH 7.2	−0.045	0.299	−0.149	0.882
	pH 7.0	0.016	0.299	0.055	0.956
	Time	0.110	0.098	1.126	0.262
	Time * [pH 7.8]	−0.115	0.139	−0.827	0.409
	Time * [pH 7.6]	−0.026	0.139	−0.188	0.851
	Time * [pH 7.4]	−0.118	0.139	−0.853	0.395
	Time * [pH 7.2]	−0.127	0.139	−0.918	0.360
	Time * [pH 7.0]	0.002	0.139	0.015	0.988

Significant effects are shown in bold. Time * pH indicates the effect of the interaction of time and pH on the angular velocity.

**Table 5 biology-12-00876-t005:** Effects of pH, time, and their interaction on highly mobile behavior in the black rockfish juveniles using GLMMs.

	Explanatory Variables	Coefficient	SE	*t* Value	*p*-Value
Highly Mobile (s)	pH 7.8	0.103	0.373	0.276	0.783
	pH 7.6	−0.192	0.373	−0.515	0.607
	pH 7.4	−0.772	0.373	−2.068	**0.040**
	pH 7.2	−1.138	0.373	−3.049	**0.003**
	pH 7.0	−0.772	0.373	−2.068	**0.040**
	Time	−0.370	0.081	−4.573	**<0.001**
	Time * [pH 7.8]	0.401	0.115	3.505	**0.001**
	Time * [pH 7.6]	0.346	0.115	3.021	**0.003**
	Time * [pH 7.4]	0.582	0.115	5.083	**<0.001**
	Time * [pH 7.2]	0.892	0.115	7.783	**<0.001**
	Time * [pH 7.0]	0.770	0.115	6.719	**<0.001**

Significant effects are shown in bold. Time * pH indicates the effect of the interaction of time and pH on the highly mobile behavior.

**Table 6 biology-12-00876-t006:** Effects of short-term exposure to pH reduction on activity state in the black rockfish juveniles.

	Time	pH					
		8.0	7.8	7.6	7.4	7.2	7.0
Highly mobile (s)	0 h	−0.40 ± 0.11	−0.17 ± 0.14	−0.42 ± 0.12	−0.26 ± 0.08	−0.93 ± 0.30	−0.65 ± 0.27
	12 h	−0.63 ± 0.11 ^a^	0.46 ± 0.16 ^b^	−0.34 ± 0.11 ^b^	−1.21 ± 0.14 ^a^	0.51 ± 0.20 ^b^	0.68 ± 0.22 ^b^
	24 h	−0.89 ± 0.18 ^a^	0.75 ± 0.09 ^b^	−0.10 ± 0.24 ^ab^	1.09 ± 0.16 ^b^	1.06 ± 0.12 ^b^	0.38 ± 0.43 ^ab^
	48 h	−1.31 ± 0.18 ^a^	0.26 ± 0.14 ^cd^	−0.79 ± 0.18 ^ab^	−0.18 ± 0.21 ^bc^	−0.11 ± 0.20 ^bc^	0.89 ± 0.13 ^d^
	96 h	−1.93 ± 0.24 ^a^	0.05 ± 0.14 ^b^	0.03 ± 0.28 ^b^	0.27 ± 0.26 ^bc^	1.99 ± 0.16 ^d^	1.22 ± 0.30 ^cd^
Mobile (s)	0 h	0.18 ± 0.01	0.17 ± 0.01	0.17 ± 0.01	0.17 ± 0.01	0.17 ± 0.01	0.17 ± 0.01
	12 h	0.16 ± 0.01 ^a^	0.14 ± 0.01 ^b^	0.14 ± 0.01 ^b^	0.12 ± 0.01 ^c^	0.14 ± 0.01 ^b^	0.14 ± 0.01 ^b^
	24 h	0.17 ± 0.01 ^a^	0.17 ± 0.01 ^a^	0.18 ± 0.01 ^a^	0.17 ± 0.01 ^a^	0.18 ± 0.01 ^a^	0.14 ± 0.01 ^b^
	48 h	0.17 ± 0.01 ^ab^	0.15 ± 0.01 ^b^	0.20 ± 0.01 ^a^	0.15 ± 0.01 ^ab^	0.16 ± 0.01 ^ab^	0.13 ± 0.01 ^b^
	96 h	0.13 ± 0.02 ^a^	0.15 ± 0.01 ^ab^	0.20 ± 0.01 ^b^	0.13 ± 0.01 ^a^	0.14 ± 0.01 ^ab^	0.12 ± 0.02 ^a^
Immobile (s)	0 h	0.08 ± 0.12	−0.51 ± 0.12	−0.21 ± 0.04	−0.19 ± 0.06	−0.47 ± 0.43	−0.07 ± 0.20
	12 h	−0.65 ± 0.08 ^a^	−0.68 ± 0.09 ^a^	−0.54 ± 0.19 ^a^	1.57 ± 0.05 ^b^	0.73 ± 0.06 ^ab^	1.19 ± 0.06 ^b^
	24 h	−0.24 ± 0.10 ^b^	−2.01 ± 0.13 ^d^	−1.35 ± 0.08 ^c^	−1.37 ± 0.13 ^c^	−1.03 ± 0.11 ^c^	0.36 ± 0.05 ^a^
	48 h	1.27 ± 0.06 ^a^	−1.02 ± 0.05 ^b^	−0.23 ± 0.08 ^b^	0.28 ± 0.15 ^ab^	0.39 ± 0.13 ^ab^	0.14 ± 0.13 ^ab^
	96 h	2.19 ± 0.11 ^a^	0.93 ± 0.19 ^b^	0.47 ± 0.09 ^b^	0.96 ± 0.06 ^b^	0.57 ± 0.11 ^b^	−0.56 ± 0.22 ^c^

Different letters indicate significant differences between groups within the same row, and same letters indicate no significant difference between groups within the same row. The values indicate estimates ± standard error using GLMMs.

**Table 7 biology-12-00876-t007:** Effects of pH, time, and their interaction on mobile behavior in the black rockfish juveniles using GLMMs.

	Explanatory Variables	Coefficient	SE	*t* Value	*p*-Value
Mobile (s)	pH 7.8	−0.009	0.012	−0.769	0.443
	pH 7.6	−0.019	0.012	−1.529	0.128
	pH 7.4	−0.004	0.012	−0.298	0.766
	pH 7.2	−0.009	0.012	−0.768	0.444
	pH 7.0	0.005	0.012	0.436	0.663
	Time	−0.004	0.004	−1.065	0.288
	Time * [pH 7.8]	0.001	0.006	0.113	0.910
	Time * [pH 7.6]	0.010	0.006	1.756	0.080
	Time * [pH 7.4]	−0.004	0.006	−0.685	0.494
	Time * [pH 7.2]	0.004	0.006	0.619	0.537
	Time * [pH 7.0]	−0.007	0.006	−1.303	0.194

Significant effects are shown in bold. Time * pH indicates the effect of the interaction of time and pH on the mobile behavior.

**Table 8 biology-12-00876-t008:** Effects of pH, time, and their interaction on immobile behavior in the black rockfish juveniles.

	Explanatory Variables	Coefficient	SE	*t* Value	*p*-Value
Immobile (s)	pH 7.8	−0.102	0.431	−0.237	0.813
	pH 7.6	0.444	0.431	1.031	0.304
	pH 7.4	1.267	0.431	2.939	**0.004**
	pH 7.2	0.831	0.431	1.926	0.055
	pH 7.0	2.137	0.431	4.956	**<0.001**
	Time	0.614	0.092	6.681	**<0.001**
	Time * [pH 7.8]	−0.361	0.130	−2.775	**0.006**
	Time * [pH 7.6]	−0.448	0.130	−3.445	**0.001**
	Time * [pH 7.4]	−0.514	0.130	−3.955	**<0.001**
	Time * [pH 7.2]	−0.440	0.130	−3.384	**0.001**
	Time * [pH 7.0]	−0.818	0.130	−6.290	**<0.001**

Significant effects are shown in bold. Time * pH indicates the effect of the interaction of time and pH on the immobile behavior.

**Table 9 biology-12-00876-t009:** Effects of pH, time, and their interaction on proximity in the black rockfish juveniles.

	Explanatory Variables	Coefficient	SE	*t* Value	*p*-Value
Proximity (s)	pH 7.8	−0.002	0.382	−0.005	0.996
	pH 7.6	−0.557	0.382	−1.457	0.146
	pH 7.4	0.297	0.382	0.776	0.438
	pH 7.2	0.108	0.382	0.283	0.777
	pH 7.0	0.183	0.382	0.479	0.632
	Time	0.462	0.082	5.666	**<0.001**
	Time * [pH 7.8]	−0.053	0.115	−0.460	0.646
	Time * [pH 7.6]	0.162	0.115	1.407	0.161
	Time * [pH 7.4]	0.043	0.115	0.369	0.712
	Time * [pH 7.2]	−0.213	0.115	−1.853	0.065
	Time * [pH 7.0]	−0.117	0.115	−1.013	0.312

Significant effects are shown in bold. Time * pH indicates the effect of the interaction of time and pH on the proximity.

**Table 10 biology-12-00876-t010:** Effects of short-term exposure to pH reduction on proximity in the black rockfish juveniles.

	Time	pH					
		8.0	7.8	7.6	7.4	7.2	7.0
Proximity (s)	0 h	−1.22 ± 0.16	−1.63 ± 0.17	−1.54 ± 0.23	−1.41 ± 0.14	−1.12 ± 0.08	−1.44 ± 0.11
	12 h	−0.13 ± 0.04 ^a^	0.92 ± 0.08 ^ab^	−0.30 ± 0.11 ^a^	1.82 ± 0.29 ^b^	−0.31 ± 0.06 ^a^	0.64 ± 0.13 ^ab^
	24 h	0.31 ± 0.02 ^a^	−0.65 ± 0.07 ^c^	−0.15 ± 0.06 ^b^	−0.54 ± 0.06 ^c^	−0.55 ± 0.05 ^c^	−0.26 ± 0.10 ^b^
	48 h	0.85 ± 0.13 ^ab^	0.23 ± 0.05 ^ab^	1.29 ± 0.07 ^a^	1.29 ± 0.20 ^a^	−0.47 ± 0.10 ^b^	0.08 ± 0.08 ^b^
	96 h	0.61 ± 0.11 ^a^	0.76 ± 0.11 ^ab^	0.78 ± 0.16 ^ab^	1.38 ± 0.21 ^b^	0.21 ± 0.11 ^a^	0.56 ± 0.19 ^a^

Different letters indicate significant differences between groups within the same row, and same letters indicate no significant difference between groups within the same row. The values indicate estimates ± standard error using GLMMs.

**Table 11 biology-12-00876-t011:** Effects of pH, time, and their interaction on immune function in the black rockfish juveniles.

	Variables	df	MS	*F* Value	*p*-Value
T-SOD	pH	5	256.100	19.179	**<0.001**
	Time	4	349.213	26.152	**<0.001**
	Time * pH	20	99.461	7.449	**<0.001**
CAT	pH	5	46004.055	3.761	**0.005**
	Time	4	16250.489	1.329	0.270
	Time * pH	20	22926.364	1.874	**0.032**
GSH-PX	pH	5	16.125	5.195	**0.001**
	Time	4	24.356	7.847	**<0.001**
	Time * pH	20	10.179	3.279	**<0.001**
AKP	pH	5	221.015	75.589	**<0.001**
	Time	4	152.130	52.030	**<0.001**
	Time * pH	20	172.233	58.905	**<0.001**

Significant effects are shown in bold. Time * pH indicates the effect of the interaction of time and pH on the immune function.

**Table 12 biology-12-00876-t012:** Effects of pH, time, and their interaction on metabolic responses in the black rockfish juveniles.

	Variables	df	MS	*F* Value	*p*-Value
GLU	pH	5	233.134	15.848	**<0.001**
	Time	4	417.010	28.348	**<0.001**
	Time * pH	20	87.558	5.952	**<0.001**
TG	pH	5	0.438	9.659	**<0.001**
	Time	4	0.883	19.480	**<0.001**
	Time * pH	20	0.175	3.872	**<0.001**

Significant effects are shown in bold. Time * pH indicates the effect of the interaction of time and pH on the metabolic responses.

**Table 13 biology-12-00876-t013:** Effects of pH, time, and their interaction on stress response in the black rockfish juveniles.

Variables	df	MS	*F* Value	*p*-Value
pH	5	0.283	2.526	**0.039**
Time	4	1.682	14.996	**<0.001**
Time * pH	20	0.553	4.927	**<0.001**

Significant effects are shown in bold. Time * pH indicates the effect of the interaction of time and pH on the stress response.

## Data Availability

The data required to reproduce these findings are available upon request by contact with the corresponding author.

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
