# Peer review of "The Effects of Short-Term Exposure to pH Reduction on the Behavioral and Physiological Parameters of Juvenile Black Rockfish (Sebastes schlegelii)"

_biology, 2023, doi:10.3390/biology12060876_

Round 1

Reviewer 1 Report

This study conducted experiments lasting four days to investigate the effects of short-term pH reduction on the behavioral and physiological responses of juvenile black rockfish. Activity state and lipid and carbohydrate metabolism were found to be affected by the pH conditions. To meet the increasing demand of the black rockfish and investigating their responses to ocean acidification, it will be better if long-term experiments can be adopted and more interesting results will be obtained. Some of the results and discussion part were not very clear to me. Comments have been marked in the attached PDF.

Reviewer 2 Report

The study highlights the subsequent impact of ocean acidifications on behavioral and physiological response of Juvenile Black rockfish which describes a conservation strategy of one of the oceanic fish resource. I went through the MS and found several loopholes and difficult statements which need clarification from the authors. Please address the following queries for more clarity.

Abstract

It’s very weak. This section is of top priority as to have a quick glance of the MS. Authors did not include the experimental fish size, design of experiment which reflects a casual approach. Results are not concrete and follow a sequence.

Introduction

The section is also not satisfactory. It composed mostly the narration of oceanic acidification process. But how it has impacted the marine organisms and what are the evidences based on earlier works. What physiological functions are mostly affected? The selection of the species here i.e. Black rockfish is based on what? What are the timeline change in population, is there any relevant data published elsewhere? What is the current oceanic pH where the species thrives? The experimental design focus on a pH above 7, which seems questionable.

English language is very poor, and redundant phrases are many.

Methods

What photoperiod regime was maintained?

What was the experimental period? Why fishes were fasted all through the experiment? This should be clear and well mentioned in abstract.

I understand that the study was based on acute toxicity, but how pH above 7 induces toxicity and whether physiological alterations have been noticed in any other test fishes?

Behavioral analysis is not well defined. Cite reference for the scores given. Further, the sampling procedure and measurement of enzymes is not clear.

Line 147: Expand “SPSS”

Methods

Table lacks units of the studied parameters

Conclusion

Line 388-389: If this is the final conclusions then several questions arise, on whether the assumed pH range is suitable to arrive at a solid conclusion.

Round 2

Reviewer 1 Report

The paper has been extensively revised. After a few minor revisions, it should be ready to be published.

Grammatical error in Line 410.

In table 5, 11, 12, etc., I guess the some of the p-value are very low instead of 0. Therefore, it is better to present them in another way, like p<0.001, or other better ways.

Author Response

The paper has been extensively revised. After a few minor revisions, it should be ready to be published.

Response: Thank you very much for reviewing our paper and for your feedback. We are pleased to hear that the revisions we made have improved the paper and brought it closer to publication. We appreciate the time and effort in reviewing our work and we will carefully consider any further comments or suggestions you may have for us. Once again, thank you for your valuable feedback.

Grammatical error in Line 410.

Response: Thank you for bringing to our attention the grammatical error in Line 410. The sentence has been modified in the revised manuscript as Lines 408-410: “However, environmental changes can increase the metabolic cost of fish, leading them to enhance their carbohydrate and lipid metabolism to cope with the high energy demands during acidification stress.”

We have since revised the sentence and thoroughly reviewed the grammar throughout the manuscript to ensure accuracy and clarity. We appreciate your attention to detail and are committed to upholding the high standards of scientific writing in our publication.

In table 5, 11, 12, etc., I guess the some of the p-value are very low instead of 0. Therefore, it is better to present them in another way, like p<0.001, or other better ways.

Response: Thank you for bringing to our attention the presentation of p-value. We have carefully reviewed and revised the tables to ensure that all irregular p-values have been modified to “< 0.001”, in accordance with standard practices in statistical analysis and reporting.
